# Distribution of Yeast Species and Risk Factors of Oral Colonization after Oral-Care Education among the Residents of Nursing Homes

**DOI:** 10.3390/jof8030310

**Published:** 2022-03-17

**Authors:** Ming-Gene Tu, Chih-Chao Lin, Ya-Ting Chiang, Zi-Li Zhou, Li-Yun Hsieh, Kai-Ting Chen, Yin-Zhi Chen, Wen-Chi Cheng, Hsiu-Jung Lo

**Affiliations:** 1School of Dentistry, China Medical University, Taichung 404, Taiwan; mgtu@mail.cmu.edu.tw (M.-G.T.); cytcyc@gmail.com (Y.-T.C.); 2Taiwan Mycology Reference Center, National Institute of Infectious Diseases and Vaccinology, National Health Research Institutes, Miaoli County 350, Taiwan; as0210639@gat.sinica.edu.tw (C.-C.L.); vitaminzzl@gmail.com (Z.-L.Z.); 980310@nhri.org.tw (L.-Y.H.); keroppi168@gmail.com (K.-T.C.); 010315@nhri.org.tw (Y.-Z.C.); 3Institute of Molecular Medicine and Bioengineering, National Yang Ming Chiao Tung University, Hsinchu 300, Taiwan; 4General Biologicals Corporation, Hsinchu 300, Taiwan; wccheng888@gmail.com; 5Department of Biological Science and Technology, National Yang Ming Chiao Tung University, Hsinchu 300, Taiwan

**Keywords:** aging, denture wearer, dry mouth, elderly, yeast colonization, oral-care education

## Abstract

Most yeasts causing infections in humans are part of commensal microflora and etiological agents of different infections when hosts become susceptible, usually due to becoming immunocompromised. The colonization of potentially pathogenic microbes in the oral cavity is increased by poor oral hygiene. This follow-up survey was conducted approximately two months after providing information on proper oral care at 10 nursing homes in Taiwan. Among the 117 of 165 residents colonized by yeasts, 67 were colonized by more than one yeast species. A total of 231 isolates comprising eight fungal genera and 25 species were identified. *Candida albicans* (44.6%) was the dominant species, followed by *Candida glabrata* (17.7%), *Candida parapsilosis* (8.7%), *Candida tropicalis* (7.8%), and *Candida pararugosa* (7.3%). Residents having a yeast colony-forming unit >10 (OR, 8.897; 95% CI 2.972–26.634; *p* < 0.001) or using a wheelchair (OR, 4.682; 95% CI 1.599–13.705; *p* = 0.005) were more likely to be colonized by multiple species. By comparing before and after oral-care education, dry mouth (OR, 3.199; 95% CI 1.448–7.068; *p* = 0.011) and having heart disease (OR, 2.681; 95% CI 1.068–6.732; *p* = 0.036) emerged as two independent risk factors for increased density of colonizing yeast.

## 1. Introduction

Poor oral hygiene contributes to the increased prevalence of the colonization of potentially pathogenic microbes including bacteria and/or fungi in the oral cavity [1]. Worldwide, fungal infections have been associated with approximately 1.5 million deaths per year [2,3]. An increase in the size of the immunocompromised population including people receiving chemotherapy, those infected with human immunodeficiency virus (HIV) or SARS-CoV-2, and the elderly has led to a significant increase in the prevalence of invasive fungal infections [4,5,6,7]. Among fungi, *Candida* species are the most frequently isolated opportunistic fungal pathogens, causing morbidity and mortality in patients. Commensal microflora including fungi in relative healthy humans are usually the etiological agents causing infections when hosts become immunocompromised [8,9]. Thus, colonization with *Candida* species is a risk for developing invasive candidiasis [10].

To monitor the trends in species distribution of yeasts causing infections in patients, we conducted a national survey in Taiwan, the Taiwan Surveillance of Antimicrobial Resistance of Yeasts. We previously found that aging is one of the risks for candiduria [11]. Impairment of the immune system, chronic diseases, medication use, poor oral hygiene, and reduced salivary flow have been identified as risk factors for Candida infections among the elderly [12]. Approximately 67% to 78% of the elderly population, average age ranging from 69.2 to 85.8 years, is colonized by yeasts [13,14,15,16]. Furthermore, aging is also a risk for oral yeast colonization/infection by more than one species [17].

Elderly including residents of nursing homes have a high degree of dental/denture plaque, which is a reservoir of pathogenic microbes [18]. The rate of detection of *Candida albicans* (66.7%) in gargled samples from elderly patients was similar to that of *Streptococcus pneumoniae* (63%), but higher than those of methicillin-resistant *Staphylococcus aureus* (14.8%) and *Pseudomonas aeruginosa* (5.6%) [19]. Moreover, in Taiwan, approximately 78% of residents in nursing homes (approximately 80% of whom have one or more chronic diseases) were colonized by yeasts [16]. In addition, the densities of yeast cells in the elderly patients were significantly higher than those in the healthy young group [19,20]. In our previous survey, we found that among the 158 of 204 residents colonized by yeasts, 88 were colonized by more than one species [16]. As expected, *C. albicans* (40.3%) was the dominant species, followed by *Candida glabrata* (15.9%), *Candida parapsilosis* (10.6%), *Candida tropicalis* (6.9%), and *Trichosporon asahii* (5.6%). Being colonized by more than 10 colony-forming unit (CFU) of yeast and being older than 70 years were independent factors associated with multi-species colonization [16].

We identified the basal line of species distribution and risk factors associated with yeast colonization and multiple species colonization among the elderly. How to diminish colonization by yeasts of these risk groups deserves to be further investigated. Thus, in the present study, we determined the effects of oral-care education on the distribution of yeast species and the risk factors for yeast colonization among residents at nursing homes in Taiwan.

## 2. Materials and Methods

### 2.1. Study and Data Collection

To evaluate the effect of the oral health education provided during the first survey, we conducted a follow-up survey approximately two months after delivering information on proper oral care. A total of 39 patients were excluded due to death or to transfers back home or to another nursing home. Thus, of the 204 residents enrolled in the previous survey [16], 165 volunteers elder than 50 years old participating in both surveys at 10 nursing homes in central Taiwan were enrolled in this follow-up survey in December 2015 after informed consents were obtained. We did not screen for the oral health of the residents prior to enrolment. At the time of collection, an oral pathologist examined the oral health of all individuals. A standardized data-collection form was used to retrieve demographic characteristics (e.g., age, sex, height, weight, and education), underlying medical conditions (e.g., diabetes mellitus, heart disease, and hypertension), and information related to dental care. Students at China Medical University’s School of Dentistry helped to obtain the information on the questionnaire including frequency of brushing teeth, mouthwash use, and the elderly having a feeling of a dry mouth recently. Number of missing teeth, number of fixed or removable dentures, and periodontal disease and/or tartar were diagnosed by a dentist.

### 2.2. Oral-Care Education

First, we emphasized the importance of having a routine dental checkup. Then, we educated the elderly about the correct method of brushing the teeth by showing videos and/or having small group discussions. We also shared information about the benefits of using interdental brushes, dental floss, and mouthwash to the elderly in small groups. When the dentist performed the checkups, she also told individuals how to take good care of their teeth. A dental care kit including a toothbrush, interdental brushes, dental floss, and mouthwash was provided to the participating elderly.

### 2.3. Sample Collection and Fungal Cultures

The sample collection was as described in the previous report [16]. Approximately 20 mL of oral rinse containing saline was obtained from each resident in December 2015. The rinses were then centrifuged and re-suspended in 1 mL of 0.85% NaCl. A total of 0.05 mL of suspension of each sample was streaked onto CHROMagar Candida media (CHROMagar, Paris, France) [21]. All medium plates were incubated at 35 °C for two days. If present, colonies from each medium plate were selected for further analyses. Additional colonies were selected from the medium plates when more than one morphotype was present. One isolate of each species from each resident was analyzed.

### 2.4. Identification

All isolates were subjected to matrix-assisted laser desorption ionization-time of flight mass spectrometry (MALDI-TOF MS) for species identification using MALDI Biotyper software (version 3.1) [22]. When isolate identifications were inconsistent with the color on CHROMagar Candida agar medium or when uncommon species were reported, the sequences of the internal transcribed spacer (ITS) region and/or the D1/D2 region of ribosomal DNA were used for species identification. The ITS regions were amplified by the primers ITS1, 5′-TCCGTAGGTGAACCTGCGG-3, and ITS4 5′-TCCTCCGCTTATTGATATGC-3′; and the D1/D2 regions were amplified by the primers NL1 5′-GCATATCAATAAGCGGAGGAAAAG-3′ and NL4 5′-GGTCCGTGTTTCAAGACGG-3′ [23].

### 2.5. Statistical Methods

SPSS software for Windows (version 12.0) was used to analyze the data. Items on the data-collection form were tested for association with frequency (incidence) of yeast colonization and multiplicity of species. Many factors were assessed including age, sex, body mass index (BMI), education, wheelchair use (mobility effect), smoking, betel nut chewing, number of missing teeth, number of dentures, frequency of brushing teeth, mouthwash use, having dry mouth, having periodontal disease and/or tartar, having chronic disease (hypertension, diabetes mellitus, heart disease, osteoporosis, stroke, gout, arthritis, kidney disease), and taking medicine. The chi-squared or Fisher’s exact test with 1-tailed correction was applied for categorical variables and the Student’s *t*-test was used for continuous variables. Logistic regression was applied to assess the independent effects of factors with values less than 0.05 in univariate analysis. A *p* value less than 0.05 was considered significant.

## 3. Results

### 3.1. Study Population

Of the 204 residents enrolled in the previous survey [16], 165 participated in this follow-up study. The average number of participants per nursing home was 16, ranging from nine to 33. The demographic data of the participants are shown in Table 1. The average age was 76.7 years, ranging from 50 to 95 years, with 119 (72.1%) older than 70 years. No differences between men and women were observed in infection rates or types. Of the sampled population, there was an average of 10 dentures each. A total of 131 (79.4%) had chronic diseases, and 50 (30.3%) needed wheelchairs.

#### Species Distribution

A total of 231 isolates, comprising eight fungal genera and 25 species, were identified including 213 Candida; seven Trichosporon; four Saccharomyces; two each of Exophiala and Magnusiomyces; and one each of Lodderomyces, Pichia, and Rhodotorula. The colonization species were *C. albicans* (103, 44.6%), *C. glabrata* (41, 17.7%), *C. parapsilosis* (20, 8.7%), *C. tropicalis* (18, 7.8%), *Candida pararugosa* (7, 3%), *Candida guilliermondii* (5, 2.2%), *T. asahii* (5, 2.2%), *Candida dubliniensis* (4, 1.7%), *Saccharomyces cerevisiae* (4, 1.7%), *Candida intermedia* (3, 1.3%), *Candida metapsilosis* (3, 1.3%), *Candida orthopsilosis* (3, 1.3%), *Candida krusei* (1, 0.4%), and other (14, 6.1%) (Figure 1).

Of the 117 residents colonized by yeasts, 50, 32, 25, eight, and two were colonized by one, two, three, four, and five species, respectively (Figure 2, Appendix A). The majority of isolates belonged to *Candida* species, accounting for 82.8% of recovered isolates. Among the 50 residents colonized by a single species, 41 (82%) were colonized by *C. albicans*, followed by four *C. parapsilosis*, 3 *C. dubliniensis*, and one each of *C. pararugosa* and *Candida utilis*. Some species were more prevalent than others. Among the two residents colonized by five species, one was colonized by *C. albicans*, *Candida intermedia*, *Candida lusitaniae*, *C. parapsilosis*, and *Pichia aff. fermentans*; and the other was colonized by *C. albicans*, *C. krusei*, *C. orthopsilosis*, *C. pararugosa*, and *Lodderomyces elongisporus*. Among the 25 residents colonized by three species, 11, five, and three were colonized by species compositions J (*C. albicans*/*C. glabrata*/*C. tropicalis*), K (*C. albicans*/*C. parapsilosis*/other), and L (*C. albicans*/*C. glabrata*/other), respectively. Among the 32 residents colonized by two species, 20, five, and three were colonized by species compositions S (*C. albicans*/*C. glabrata*), T (*C. albicans*/*C. tropicalis*), and U (*C. albicans*/*C. parapsilosis*), respectively.

### 3.2. Status and Risk Factors for Yeast Colonization

According to the univariate analysis, the rate of yeast colonization was increased when the elderly had more dentures (*p* < 0.001), more missing teeth (*p* = 0.018), dry mouth (*p* = 0.037), periodontal diseases (*p* = 0.026), or were older (*p* = 0.002). Approximately 78% of residents colonized by yeast were older than 70 years (*p* = 0.011). Having more dentures (OR, 1.06; 95% CI 1.015–1.107; *p* = 0.008) is a risk for being colonized by yeast, according to the multivariate analysis. Furthermore, residents having yeast CFU >10 (OR, 8.897; 95% CI 2.972–26.634; *p* < 0.001) or using a wheelchair (OR, 4.682; 95% CI 1.599–13.705; *p* = 0.005) were more likely to be colonized by multiple species (Table 2). By comparing the two surveys, dry mouth (OR, 3.199; 95% CI 1.448–7.068; *p* = 0.011) and having heart diseases (OR, 2.681; 95% CI 1.068–6.732; *p* = 0.036) were found to be two independent risk factors for increased density of colonizing yeast (Table 3).

## 4. Discussion

Because of transfers or death, only 165 of the initial 204 residents [16] were in the follow-up survey. The characteristics of residents of both surveys were similar with one exception: using mouthwash. After oral-care education, significantly more (*p* < 0.001) residents used mouthwash (50.9% in the present survey vs. 25% in the previous one). Even though the difference was not statistically significant, oral-care education appeared to result in a reduced proportion of residents colonized by yeast (158/204 vs. 117/165 *p* = 0.152). It would be interesting to conduct a follow-up survey to investigate the effect of oral-care education in the long-term.

We previously used a traditional epidemiological approach to establish a baseline of yeast colonization among residents at nursing homes in Taiwan. In the present study, we found that up to 70.9% of residents at nursing homes in central Taiwan were colonized by yeast, a rate higher than the 67% reported in France [14], but lower than the 77.5% in the previous survey [16] and the 75% reported in Italy [13]. As expected, age, having dentures or missing teeth, periodontal disease, and dry mouth were risk factors for being colonized by yeast [12,16,24,25,26]. Interestingly, in our previous study, brushing teeth more than once a day was a risk factor for yeast colonization [16]. After our staff educated nursing home residents in the correct way to brush teeth, especially in having them brush their teeth gently to avoid unnecessary damage, frequency of their brushing teeth was no longer a risk factor.

Of the 25 species recovered in the present study, only four (*Candida ethanolica*, *Candida fermentati*, *P. aff. fermentans*, and *Trichosporon terricola*) have not been reported to cause diseases in humans. There were seven residents colonized by *C. pararugosa*, which was first reported in 2017 to cause bloodstream infections [27]. Hence, it is worth noting that in addition to those common species that cause diseases in humans, some rare species that were recovered in the present study such as *Candida norvegensis* [28], *C. utilis* [29], *Cystobasidium minutum* (*Rhodotorula minute*) [30], *Exophiala dermatitidis* [31,32,33], *Lodderomyces elongisporus* [34,35], *Magnusiomyces capitatus* [36], and *Trichosporon mucoides* [37] have been reported to cause invasive infections in humans.

There were fewer genera (12 vs. 8) and species (33 vs. 25) in the present survey than in the previous one. The species distribution between the two surveys was similar among *Candida* spp. The prevalence of some rare species differed across the two surveys. For example, *Candida kefyr*, *Cryptococcus neoformans var grubii*, *Fereydounia khargensis*, *Lachancea fermentati*, *Pichia manshurica*, *Rhodosporidium toruloides*, *Rhodotorula mucilaginosa*, *Trichosporon faecale*, and *Trichosporon jirovecii* were detected in the first-round survey but not in the present one. On the other hand, *P. aff*. *fermentans*, *C. minutum*, and *T. terricola* were detected only in the present survey. Interestingly, the present survey found significantly less (*p* = 0.04) *T. asahii* (5, 2.2%) than the previous one (18, 5.6%). This reduction in colonization by *T. asahii* is not due to dropout of the subjects, because 13 of the 18 residents colonized by *T. asahii* in the first study were included in the follow-up present survey. Whether increased use of mouthwash or proper oral care contributed to the reduced diversity of spp. and reduced detection of *T. asahii* needs further investigation.

Of the 266 participants older than 60 years in a study in Japan, 162 were colonized by *Candida* species in tongue dorsa. *Candida albicans*, *C. glabrata*, *C. tropicalis*, and *C. parapsilosis* were detected in 142, 60, five, and two participants, respectively. No *C. krusei* was detected in that survey [25]. Interestingly, the species distribution in the Japanese study was different from that in Taiwan. Oral yeast colonization of 323 healthy individuals with a mean age of 33.9 years (range 18–60) was previously studied. Among the 52 isolates, comprising 13 species, 57.7% were *C. albicans*, 15.4% were *C. parapsilosis*, and 3.9% were *C. glabrata*. No *C. tropicalis* was detected in that study [20]. In the 2018 Taiwan Surveillance of Antimicrobial Resistance of Yeasts, 34%, 25.8%, 24.8%, 7%, and 1.7% of 1266 isolates from patients were *C. albicans*, *C. glabrata*, *C. tropicalis*, *C. parapsilosis*, and *C. krusei*, respectively (D.-J.T.; K.-Y.T.; Y.-Z.C.; P.-N.C.; L.-Y.H.; H.-J.L.; TSARY Hospitals. The distribution and drug susceptibilities of clinical Candida species in TSARY 2018. In preparation.). Factors contributing to the species distribution of oral yeast colonization at nursing homes more similar to that of patients than that of the healthy young population need further investigation.

Along with chewing betel nuts and aging, having dentures, periodontal disease, and other chronic diseases are expected risk factors for being colonized by yeast [12,24,25,26]. According to the multivariate analysis, having more dentures was the only risk factor for being colonized by yeast in the nursing home residents in Taiwan before and after oral-care education surveys. There was a significant oral *C. albicans* colonization rate among denture wearers compared with non-denture wearers (61.5% vs. 33.6%). In addition, there was a significant oral carriage rate of *C. albicans* among complete denture wearers compared with partial denture wearers (73.9% vs. 51.7%) [38]. The limitation of the present study is that we did not distinguish fixed and removable dentures among residents since these may have different effects on Candida colonization. Interestingly, in the previous survey before oral-care education, brushing teeth more than once a day, having a chronic disease, and being female were identified by univariate analysis as potential risk factors for being colonized by yeasts; but in the follow-up survey, these were no longer such risks. Furthermore, in the present survey, we also identified that residents having dry mouth were more likely to be colonized by yeast. Since we did not assess compliance after oral-care education, we could not distinguish whether the differences between these two successional surveys were due to oral-care education or to increased use of mouthwash. Hence, a study could be designed to address this question.

## Figures and Tables

**Figure 1 jof-08-00310-f001:**
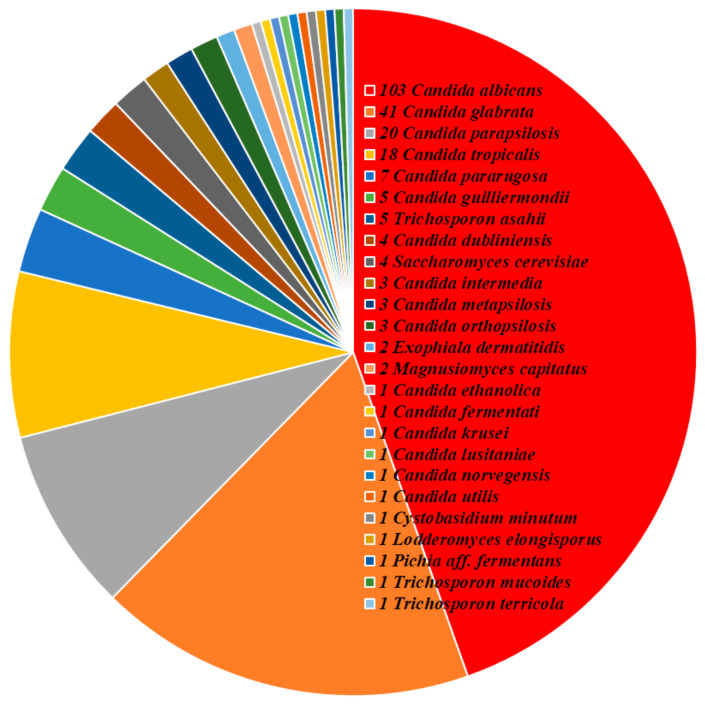
Distribution of species of 231 isolates. The number in front of the name of a species refers to the number of isolates recovered in the present study.

**Figure 2 jof-08-00310-f002:**
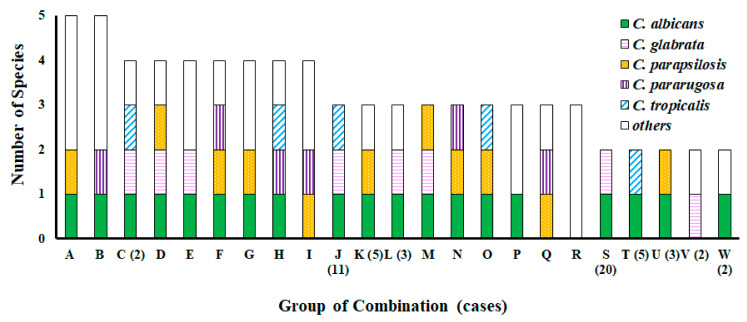
Distribution of species composition of residents colonized by multiple species. The number in parentheses refers to the number of the combination of species. Where a number is not given, there was only one case in the combination.

**Table 1 jof-08-00310-t001:** Risk factors of residents colonized by yeasts.

Characteristics	Total (N = 165)		Colonized by Yeast				Univariate	Multivariate
			yes (N = 117)		no (N = 48)			
	Mean	SD	Mean	SD	Mean	SD	*p* Value	*p* Value, OR (95% CI)
Age, years (mean ± SD)	76.7	10.26	78.3	9.61	72.8	10.84	**0.002**	
BMI	22.6	3.66	22.6	3.71	22.8	3.56	0.702	
Number of missing teeth	16.0	10.03	17.2	9.99	13.1	9.62	**0.018**	0.638, 1.01 (0.969–1.053)
Number of dentures	10.9	11.32	13.0	11.86	5.8	7.93	**<0.001**	**0.008, 1.06** **(1.015–1.107)**
	No.	%	No.	%	No.	%	*p* value	
Female	87	52.7	65	55.6	22	45.8	0.256	
Age >70 years	119	72.1	91	77.8	28	58.3	**0.011**	0.305, 1.514 (0.686–3.341)
Education							0.153	
None	25	15.2	20	17.1	5	10.4		
Elementary school	81	49.1	54	46.2	27	56.3		
Junior high school	25	15.2	22	18.8	3	6.3		
High school	20	12.1	12	10.3	8	16.7		
College or above	14	8.5	9	7.7	5	10.4		
Wheelchair use	50	30.3	32	27.4	18	37.5	0.198	
Smoking	15	9.1	14	12	1	2.1	0.053	
Betel nut chewing	6	3.6	3	2.6	3	6.3	0.147	
Frequency of brushing teeth > once a day	96	58.2	72	61.5	24	50	0.172	
Mouthwash	84	50.9	63	53.8	21	43.8	0.239	
Dry mouth	54	32.7	44	37.6	10	20.8	**0.037**	0.608, 1.218 (0.573–2.589)
Periodontal disease and/or tartar	102	61.8	66	56.4	36	75	**0.026**	0.188, 0.58 (0.258–1.304)
Chronic disease	131	79.4	97	82.9	34	70.8	0.118	
Hypertension	73	44.2	55	47	18	37.5	0.264	
Diabetes mellitus	37	22.4	31	26.5	6	12.5	0.05	
Heart disease	30	18.2	23	19.7	7	14.6	0.443	
Osteoporosis	8	4.8	8	6.8	0	0	0.106	
Stroke	11	6.7	10	8.5	1	2.1	0.179	
Gout	9	5.5	7	6	2	4.2	1	
Arthritis	8	4.8	8	6.8	0	0	0.106	
Kidney disease	2	1.2	2	1.7	0	0	1	
Taking medicine	114	69.1	84	71.8	30	62.5	0.358	

No., number.

**Table 2 jof-08-00310-t002:** Risk factors of residents colonized by multiple species.

Characteristics	Total (N = 117)		More Than One Species (N = 67)		One Species (N = 50)		Univariate	Multivariate
	Mean	SD	Mean	SD	Mean	SD	*p* Value	*p* Value, OR (95% CI)
Age	78.3	9.61	79.2	8.74	76.9	10.59	0.165	
BMI	22.6	3.71	22.5	3.55	22.7	3.96	0.757	
Number of missing teeth	17.2	9.99	17.4	10.20	16.9	9.79	0.807	
Number of dentures	13.0	11.86	14.0	11.99	11.6	11.67	0.29	
	No.	%	No.	%	No.	%	*p* value	
Yeast CFU >10	90	76.9	61	91	29	58	**<0.001**	**<0.001, 8.897 (2.972–26.634)**
Female	65	55.6	38	56.7	27	54	0.77	
Age >70	91	77.8	55	82.1	36	72	0.194	
Education							0.323	
None	20	17.1	9	13.4	11	22		
Elementary school	54	46.2	32	47.8	22	44		
Junior high school	22	18.8	16	23.9	6	12		
High school	12	10.3	5	7.5	7	14		
College or above	9	7.7	5	7.5	4	8		
Wheelchair use	32	27.4	25	37.3	7	14	0.005	**0.005, 4.682 (1.599–13.705)**
Smoking	14	12	9	13.4	5	10	0.871	
Betel nut chewing	3	2.6	2	3	1	2	1	
Frequency of brushing teeth > once a day	72	61.5	40	59.7	32	64	0.636	
Mouthwash	63	53.8	37	55.2	26	52	0.729	
Dry mouth	44	37.6	28	41.8	16	32	0.279	
Periodontal disease and/or tartar	66	56.4	35	52.2	31	62	0.292	
Chronic disease	97	82.9	55	82.1	42	84	1	
Hypertension	55	47	33	49.3	22	44	0.573	
Diabetes mellitus	31	26.5	18	26.9	13	26	0.916	
Heart disease	23	19.7	14	20.9	9	18	0.697	
Osteoporosis	8	6.8	6	9	2	4	0.464	
Arthritis	8	6.8	5	7.5	3	6	1	
Stroke	10	8.5	5	7.5	5	10	0.742	
Gout	7	6	4	6	3	6	1	
Kidney disease	2	1.7	0	0	2	4	0.181	
Taking medicine	84	71.8	51	76.1	33	66	0.209	

No., number.

**Table 3 jof-08-00310-t003:** Risk factors for higher density of colonizing yeasts.

	Total (N = 165)		More CFU (N = 34)		Less/Same CFU (N = 131)		Univariate	Multivariate
	N/Mean	%/SD	N/Mean	%/SD	N/Mean	%/SD	*p* Value	*p* Value, OR (95% CI)
Age	76.7	10.26	75.06	9.313	77.12	10.48	0.297	
BMI	22.63	3.66	22.34	3.191	22.7	3.78	0.607	
Number of missing teeth	15.98	10.03	14.15	9.372	16.46	10.17	0.232	
Number of dentures	10.87	11.32	11.97	11.04	10.59	11.41	0.527	

Female	87	52.7	18	52.9	69	52.7	0.978	
Age >70	119	72.1	26	76.5	93	71	0.526	
Education							0.173	
None	25	15.2	8	23.5	17	13		
Elementary	81	49.1	13	38.2	68	51.9		
Junior high	25	15.2	8	23.5	17	13		
High school	20	12.1	2	5.9	18	13.7		
College or above	14	8.5	3	8.8	11	8.4		
Wheelchair use	50	30.3	12	35.3	38	29	0.477	
Smoking	15	9.1	4	11.8	11	8.4	0.761	
Betel nut chewing	6	3.6	2	5.9	4	3.1	0.686	
1st frequency of brushing teeth >1/day	88	53.3	15	44.1	73	55.7	0.227	
1st mouthwash	41	24.8	6	17.6	35	26.7	0.275	
1st dry mouth	61	37	10	29.4	51	38.9	0.306	
2nd frequency of brushing teeth >1/day	95	57.6	18	52.9	77	58.8	0.798	
2nd frequency of using floss >1/day	30	18.2	8	23.5	22	16.8	0.657	
2nd mouthwash	84	50.9	19	55.9	65	49.6	0.515	
2nd mouth spread >1/day	25	15.2	8	23.5	17	13	0.267	
2nd dry mouth	54	32.7	19	55.9	35	26.7	**0.003**	**0.011, 2.816** **(1.265–6.268)**
1st periodontal disease and/or tartar	102	61.8	21	61.8	81	61.8	0.994	
Seeing dentist after 1st survey	33	20	6	17.6	27	20.6	0.924	
Chronic disease	131	79.4	29	85.3	102	77.9	0.584	
Hypertension	73	44.2	16	47.1	57	43.5	0.711	
Diabetes mellitus	37	22.4	10	29.4	27	20.6	0.273	
Heart disease	30	18.2	11	32.4	19	14.5	**0.016**	**0.036, 2.681** **(1.068–6.732)**
Osteoporosis	9	5.5	1	2.9	8	6.1	0.687	
Stroke	11	6.7	4	11.8	7	5.3	0.24	
Gout	8	4.8	3	8.8	5	3.8	0.363	
Arthritis	8	4.8	2	5.9	6	4.6	0.669	
Kidney disease	2	1.2	1	2.9	1	0.8	0.371	
Taking medicine	114	69.1	25	73.5	89	67.9	0.798	

No., number; CFU, colony-forming unit.

## Data Availability

Not applicable.

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
