# Peer review of "Distribution of Yeast Species and Risk Factors of Oral Colonization after Oral-Care Education among the Residents of Nursing Homes"

_jof, 2022, doi:10.3390/jof8030310_

Round 1

Reviewer 1 Report

The manuscript presents information that might interest the readers. However, several correction and clarification are required to improve the manuscript.

  1. The objective of the study was to evaluate effects of oral care education on Candida colonization species. However, the authors compared the results with previous article published elsewhere. It would be better to re-analyze the data by excluding the ones that dropped out during the study, and present before and after data in the current manuscript.
  2. Inclusion and exclusion criteria of the study population should be explained. Criteria in evaluating dry mouth, periodontal disease, etc. should be stated.
  3. Because the main goal of the study was to evaluate the effects of oral care education, detail of oral care education provided should be explained. Is there any method to assess the compliance of the subjects in oral care? If applicable, the analysis could be separated into well-compliance group vs non-compliance group.
  4. Many factors are related and could affect the statistical analysis results. Multivariate analysis is more appropriate than univariate. The data interpretation should depend on multivariate analysis data.
  5. If the data is re-analyzed as suggested, statistical analysis would be changed accordingly.
  6. Please explain “number of denture”. Did the authors consider fixed or removable or both types of denture?
  7. Minor grammatical errors are observed. Please revise. Moreover, many sentences were similar to the previously published manuscript (reference 15), please re-write the sentence.

Reviewer 2 Report

the concept of the article is interesting but the manuscript has to be drafted properly for scientific soundness. comments are highlighted in the document below

Round 2

Reviewer 1 Report

  1. The authors still have not stated inclusion and exclusion criteria of the study population. For example, residents in nursing homes during year …
  2. Limitation of the study should be stated regarding the lack of compliance assessment.
  3. Fixed and removable dentures may have different effects on Candida colonization. This aspect should be included in the discussion.

Reviewer 2 Report

the manuscript is well-drafted and all the comments given before are addressed

Author Response

Thank you for your time and suggestions to help us to improve our manuscript.

The English language spell have been checked.